# Analysis of a Real-World Cohort of Metastatic Breast Cancer Patients Shows Circulating Tumor Cell Clusters (CTC-clusters) as Predictors of Patient Outcomes

**DOI:** 10.3390/cancers12051111

**Published:** 2020-04-29

**Authors:** Clotilde Costa, Laura Muinelo-Romay, Victor Cebey-López, Thais Pereira-Veiga, Inés Martínez-Pena, Manuel Abreu, Alicia Abalo, Ramón M. Lago-Lestón, Carmen Abuín, Patricia Palacios, Juan Cueva, Roberto Piñeiro, Rafael López-López

**Affiliations:** 1Roche-Chus Joint Unit, Translational Medical Oncology Group, Oncomet, Health Research Institute of Santiago de Compostela (IDIS), Travesía da Choupana s/n, 15706 Santiago de Compostela, Spain; clotilde.costa.nogueira@sergas.es (C.C.); thaispv85@gmail.com (T.P.-V.); ines.martinez.pena@rai.usc.es (I.M.-P.); carmen.abuin.redondo@sergas.es (C.A.); rafael.lopez.lopez@sergas.es (R.L.-L.); 2CIBERONC, Centro de Investigación Biomédica en Red Cáncer, 28029 Madrid, Spain; laura.muinelo.romay@sergas.es; 3Liquid Biopsy Analysis Unit, Translational Medical Oncology Group, Health Research Institute of Santiago de Santiago de Compostela (IDIS), Travesía da Choupana s/n, 15706 Santiago de Compostela, Spain; maabreu20@gmail.com (M.A.); alicia.abalo.pineiro@sergas.es (A.A.); ramon.manuel.lago.leston@sergas.es (R.M.L.-L.); 4Department of Oncology, Complexo Hospitalario Universitario de Santiago de Compostela (SERGAS), 15706 Santiago de Compostela, Spain; victor.cebey.lopez@sergas.es (V.C.-L.); patricia.palacios.ozores@sergas.es (P.P.); juan.fernando.cueva.banuelos@sergas.es (J.C.)

**Keywords:** breast cancer, metastasis, circulating tumor cells (CTCs), CTC-clusters, prognosis

## Abstract

Circulating tumor cell (CTC) enumeration has emerged as a powerful biomarker for the assessment of prognosis and the response to treatment in metastatic breast cancer (MBC). Moreover, clinical evidences show that CTC-cluster counts add prognostic information to CTC enumeration, however, their significance is not well understood, and more clinical evidences are needed. We aim to evaluate the prognostic value of longitudinally collected single CTCs and CTC-clusters in a heterogeneous real-world cohort of 54 MBC patients. Blood samples were longitudinally collected at baseline and follow up. CTC and CTC-cluster enumeration was performed using the CellSearch^®^ system. Associations with progression-free survival (PFS) and overall survival (OS) were evaluated using Cox proportional hazards modelling. Elevated CTC counts and CTC-clusters at baseline were significantly associated with a shorter survival time. In joint analysis, patients with high CTC counts and CTC-cluster at baseline were at a higher risk of progression and death, and longitudinal analysis showed that patients with CTC-clusters had significantly shorter survival compared to patients without clusters. Moreover, patients with CTC-cluster of a larger size were at a higher risk of death. A longitudinal analysis of a real-world cohort of MBC patients indicates that CTC-clusters analysis provides additional prognostic value to single CTC enumeration, and that CTC-cluster size correlates with patient outcome.

## 1. Introduction

Breast cancer (BC) corresponds to the most frequent tumor type in women [1]. Nearly 30% of women initially diagnosed with early-stage BC will go on to develop metastatic lesions [2]. Disease progression and the appearance of a disseminated disease have a negative impact on the survival of these patients, being metastases responsible for 90% of deaths related to cancer [3]. Despite the positive impact of new systemic therapies incorporated to the clinic in recent years, metastatic breast cancer (MBC) remains an incurable disease. Currently, treatment selection is mainly based on the histological characterization of a biopsy from the primary tumor or metastatic sites. However, tissue biopsies have inherently associated limitations, such as restricted access and lack of representation of intratumoral heterogeneity, making them an unfeasible approach for long term disease monitoring [4]. Therefore, less invasive techniques, with the capacity to predict patient clinical outcome and to guide the treatment are needed, in order to better select the most convenient therapy.

Circulating tumor cells (CTCs), those cells shed into the blood stream by both primary and metastatic tumors, are the main responsible for metastases formation at distant sites. CTCs therefore have the potential to serve as a liquid biopsy. The analysis of CTCs represents a non-invasive procedure that can be serially repeated in order to assess tumor progression in real time, and CTC enumeration may hold promise for improving cancer treatment. CTCs are frequently detected in the peripheral blood of patients with MBC, but rarely found in early BC [5,6,7]. The prognostic value of CTCs in MBC is extensively studied and has been corroborated by large studies. Thus, a CTC count of ≥5 cells per 7.5 mL blood detected by the CellSearch^®^ system is an independent predictive factor of worse progression-free survival (PFS) and overall survival (OS). The first evidence in this regard was shown in 2004 [8], and since then, many studies have confirmed these findings [5,9,10,11,12]. The highest level of evidence has been shown in a recent pooled analysis of individual patient data gathered from 1944 patients, demonstrating the clinical validity of CTC enumeration, and that CTC count improves the prognostication of MBC [5]. Moreover, other studies indicate that CTC dynamics seem to reflect treatment response, and that CTC status can serve as an indicator to monitor the effectiveness of treatments and guide subsequent therapies in BC [5,13,14,15].

CTCs can be found in the bloodstream of cancer patients as single cells or as aggregates known as CTC-clusters or circulating tumor microemboli [16,17,18,19,20,21]. Although rare in circulation, CTC-clusters show a higher ability to form distant metastases than single CTCs [22,23,24]. Key biological features of CTC-cluster reveal the molecular mechanisms underlying their enhanced metastatic potential, i.e., a low incidence of apoptosis, a hybrid epithelial/mesenchymal phenotype, and a stem cell signature acquired through the modification of epigenetic programs [16,17,22,25,26]. Similarly to CTCs, the enumeration of CTC-clusters in the blood of MBC patients has been revealed as a strong biomarker for the assessment of prognosis and response to treatment [27,28,29,30]. Thus, patients with CTC-clusters have significantly worse OS and PFS than those patients with only single CTCs. Recent studies have shown that CTC-cluster evaluation allows for the stratification of MBC patients with elevated baseline CTCs into different survival groups. In fact, CTC-cluster counts add prognostic value to CTC enumeration alone, and even more importantly, longitudinal evaluation of CTCs and CTC-clusters improves prognostication and treatment monitoring in patients with MBC [16,27,28,29,30]. On the other hand, recent data from a clinical trial indicates that the absolute number of CTCs, rather than the presence of CTC-clusters, are predictors of the outcome of MBC patients starting first line chemotherapy [31].

The aim of this study was to evaluate the prognostic value of CTC counts and CTC-clusters determined by CellSearch^®^ in an unselected cohort of MBC patients, and to examine how these relate to PFS and OS. For this purpose, we use longitudinal collected data at baseline and follow up from a heterogeneous cohort of patients diagnosed with MBC starting first line of therapy and MBC patients initiating a new line of therapy.

## 2. Results

### 2.1. Patient Characteristics 

A total of 54 MBC patients were included in the study. The median follow up time from baseline for alive patients was 197 days (range 22–603 days). Among the patients, 27 (50.9%) were classified as having hormone receptor–positive and human epidermal growth factor receptor 2-negative tumors (HR+HER2-); 11 (20.8%) were classified as having HER2+ tumors; and 15 (28.3%) as triple-negative breast cancer tumors (HR-HER2-). One patient showed an undetermined HER2 staining. Visceral metastases were present in 40 patients (74.1%), while 14 patients (25.9%) had non-visceral metastasis. Patient characteristics are detailed in Table 1.

### 2.2. Counts of CTCs and CTC-Clusters and Association with Clinicopathological Variables

CTCs and CTC-clusters were analyzed in a total of 96 blood samples collected at baseline (when patients started first-line of systemic therapy or before starting a new line of therapy) from 54 patients, of whom 38 patients had follow up samples collected (Appendix A). Among the patients analyzed at baseline, 31 (57.4%) had ≥5 CTCs, and 14 (25.9%) had at least one CTC-cluster (composed of ≥2 CTCs). Within the patients considered at follow up visit, 13 (34.2%) had ≥ 5 CTCs, and 7 (18.4%) had at least one CTC-cluster detected. The fraction of patients with ≥5 CTCs decreased from baseline to follow up (Appendix A). CTC-clusters were exclusively found in patients with ≥5 CTCs, 14 of 31 (45.2%) at baseline, and 7 of 13 (53.8%) at follow up, and their presence was clearly associated with the number of CTCs (Appendix A). At baseline, 12 patients presented CTC-clusters composed by two or three cells, and two patients presented CTC-clusters composed by ≥4 cells. During follow up, five patients presented CTC-clusters composed by two or three cells, and two patients presented CTC-clusters composed by ≥4 cells. At both time points, a higher proportion of patients showing ≥5 CTCs and CTC-clusters was found in the HR+HER2- subtype, however, we did not find significant differences on the distribution in relation to BC subtype at baseline and follow up (Appendix A). No association was found between the number and location of metastases with the counts of CTCs or CTC-clusters.

The presence of heterotypic clusters, defined by the detection of clustered CTCs and CD45-positive white blood cells (WBCs) (≥1 CTC and ≥1 CD45+ cell), was also evaluated (Figure 1). At baseline, two patients (3.7%) had at least one CTC-WBC cluster detected, and at follow up three patients (7.9%) had at least one CTC-WBC cluster identified (Appendix A, Appendix A). The presence of clustered CTC-WBC was associated with an elevated CTC count (≥5 CTCs/7.5 mL), both at baseline and follow up (Appendix A).

These results evidenced that CTC-clusters are found in a low frequency in patients with MBC, being clearly more often found in patients with a high spread of single CTCs, before the treatment onset and during the follow up.

### 2.3. Presence of CTC-Clusters Predicts Patient Outcome at Baseline

Patients with ≥5 CTCs/7.5 mL at baseline (*n* = 31) had a trend towards an inferior PFS (*P*_log-rank_ = 0.059), and they had a significantly shorter OS (*P*_log-rank_ = 0.017), compared to those with none or < 5 CTCs (Figure 2a,b). Accordingly, Cox regression analysis showed that patients with high CTC count (≥5 CTCs) were at a higher risk of death (HR_OS_ 3.15; 95% CI 1.16–5.55; *p* = 0.024), but not of progression (HR_PFS_ = 2.11; 95% CI 0.95–4.69; *p* = 0.065), than those with <5 CTCs (Table 2). In the multivariable analysis, both observations resulted in being significant after adjustment for other clinicopathological variables (described in Appendix A). In CTC-cluster based analysis, patients with ≥1 CTC-cluster had a significantly increased risk of disease progression (HR_PFS_ 3.95; 95% CI 1.80–8.68; *p* = 0.0006) and death (HR_OS_ 4.23; 95% CI 1.8–10.1; *p* = 0.0009), and showed shorter PFS and OS (PFS and OS *P*_log-rank_ < 0.001) (Figure 2c,d). Importantly, both observations maintained their significance when adjusting for other clinicopathological variables (Table 2). On the other hand, the low frequency presence of CTC-WBC cluster at baseline was not able to predict patient outcomes.

Patients with ≥5 CTCs (*n* = 13) at follow up did not show an increased risk of progression or shorter PFS time, but they had an increased risk of death (HR_OS_ 5.3; 95% CI 1.4–21; *p* = 0.017) and a shorter OS (*P*_log-rank_ < 0.05) compared to patients with <5 CTCs/7.5 mL (Table 2, Figure 2e,f). Cox regression analysis resulted in being not significant when adjusting for other prognostic factors. In CTC-cluster based analysis, PFS or OS were not different for patients with or without CTC-clusters, although a trend was observed towards an increased risk of death (HR 3.0) (Figure 2g,h, Table 2). Similar to the baseline analysis, no prognostic value was observed when the presence of CTC-WBC cluster was evaluated.

### 2.4. Joint Analysis of CTCs and CTC-Clusters for Patient Outcome Prediction

We investigated the joint effect of CTCs and CTC-clusters in our cohort by classifying patients in the three risk groups previously described [27]: <5 CTCs without any CTC-cluster (low-risk), ≥5 CTCs without any CTC-cluster (medium-risk), and ≥5 CTCs with CTC-clusters (high-risk). Patients within the high-risk group had a shorter PFS and OS (*P*_log-rank_ < 0.01), and an increased risk of progression and death (HR_PFS_ 4.3; 95% CI 1.76–10.6; *p* = 0.0013; HR_OS_ 5.8; 95% CI 1.96–17.1; *p* = 0.0014) compared to patients with <5 CTC (Figure 3a,b, Table 3). The group of patients from the medium-risk group did not show increased risk of progression or mortality (HR_PFS_ 1.24; 95% CI 0.47–3.24; *p* = 0.66; HR_OS_ 1.8; 95% CI 0.59–5.98; *p* = 0.28). The prognostic value of CTC-clusters remained significant in the multivariate analysis when adjusting for different clinicopathological variables, including the site of metastasis, the only significant prognostic marker in multivariable analysis in this cohort (Appendix A). These data indicate that in this cohort of patients, baseline CTC-clusters are an independent prognostic factor adding value to the enumeration of CTCs alone.

In addition to this analysis, we also applied a previously described threshold of ≥20 CTCs [30,31], and analyzed its relationship with patient outcome. In an initial analysis, we observed that patients with ≥20 CTCs at baseline were at a higher risk of death than patients with ≥5 CTCs (Appendix A). Interestingly, when we classified patients in risk groups based on the presence or absence of CTC-clusters, we found that patients with ≥20 CTCs and CTC-clusters had an increased risk of death compared with patients with <20 CTCs (HR 5.5; 95% CI 2.2–13.7; *p* = 0.0002). The prognostic value of CTC-clusters remained significant in the multivariate analysis when adjusting for different clinicopathological variables. However, when we specifically looked at the group of patients with ≥ 20 CTCs, the presence of CTC-clusters did not add prognostic information (data not shown).

When focusing on patients at follow up, we observed that patients with ≥5 CTCs without any CTC-cluster and patients with ≥5 CTCs and CTC-clusters had a shorter OS (*P*_log-rank_ < 0.05) and increased risk of death (HR_OS_ 5.71; 95% CI 1.11–29.42; *p* = 0.037, and HR_OS_ 5.0; 95% CI 1.11–22.40; *p* = 0.035, respectively) compared to patients with <5 CTC (Figure 3c,d, Table 3). However, in Cox regression analysis adjusting for different clinicopathological variables these effects were lost. No increased risk of progression was observed in either of both groups. Taken together, and in agreement with previous reports, CTC-clusters might provide independent prognostic value in patients with elevated CTCs, particularly at baseline.

### 2.5. Patient Outcome Prediction Based on Baseline-to-Follow up Changes of CTCs and CTC-Clusters

To evaluate CTCs and CTC-clusters as early predictors of progression, changes in CTC and CTC-cluster counts from baseline to follow up in relation to clinical outcomes were analyzed on the 38 patients, from whom samples were collected at both time points. Regarding CTC analysis, patients were classified in four groups: (i) <5 CTCs both at baseline and follow up; (ii) ≥5 CTCs at baseline and <5 CTCs at follow up; (iii) <5 CTCs at baseline and ≥5 CTCs at follow up; and (iv) ≥5 CTCs both at baseline and follow up. Only one patient from group iii (<5 CTCs at baseline and ≥ 5 CTCs at follow up) was found in this cohort, and it was not included in the posterior analysis. Changes in CTC counts from baseline to follow up were not able to predict patient outcomes in terms of PFS, although a significant effect was observed in terms of OS. Patients with ≥ 5 CTCs, both at baseline and follow up, showed a significantly shorter OS (*P*_log-rank_ = 0.037), and a higher risk of death, with a HR of 3.9 (95% CI 0.97–15.8; *p* = 0.055) (Figure 4, Table 4). 

With regard to CTC-cluster analysis, patients were classified into two risk groups: i) low risk, comprised of patients without CTC-clusters at both time points or with a reduction in the number from baseline to follow up; and ii) high risk, comprised of patients whose CTC-cluster count did not change or increase over time. Changes in CTC-cluster counts from baseline to follow up were not able to predict patient outcome in terms of PFS and OS, however, for patients in the high risk group, there was a clear trend towards a shorter OS (*P*_log-rank_ = 0.07) and increased risk of death, with a HR of 2.96 (95% CI 0.83–10.5; *p* = 0.09) (Figure 4, Table 4).

### 2.6. Longitudinal Changes of CTC-Clusters and Cluster Size Predict Patient Outcomes

We took advantage of the 96 samples longitudinally collected from all MBC patients included in the study, to perform a time dependent analysis regarding CTC-cluster data. For this analysis, we initially stratified patients in four risk groups combining CTC counts and the presence of CTC-clusters: patients without CTCs, patients with 1–4 CTC without CTC-clusters, patients with ≥5 CTCs without CTC-clusters, and patients with ≥5 CTCs with CTC-clusters. This analysis revealed that patients with ≥5 CTCs with ≥1 CTC-cluster had shorter PFS and OS (*P*_log-rank_ < 0.01; Figure 5a,b). Indeed, patients with ≥ 5 CTCs with ≥1 CTC-cluster had a HR for PFS and OS of 4.65 (*p* = 0.002) and 13.9 (*p* = 0.0004), respectively, compared to those patients with no CTCs. A similar result was observed when the group of patients with 1–4 CTCs without CTC-cluster was taken as reference (Appendix A). In contrast, patients with ≥5 CTCs without CTC-cluster only showed an increased risk of death (HR_OS_ 7.38; 95% CI 1.68–32.4; *p* = 0.008). However, this observation did not remain significant when patients with 1–4 CTCs without any CTC-cluster were taken as references.

We also analyzed the predictive value of CTC-cluster size. Patients with 2–3 cell CTC-cluster had a higher risk of disease progression (HR_PFS_ 3.9; 95% CI 1.58–9.9; *p* = 0.003) than patients without CTCs, and this increase was more pronounced for patients with ≥4 cell CTC-cluster (HR_PFS_ 4.1; 95% CI 1.05–16.0; *p* = 0.041), with a mean PFS of 45 and 56 days, respectively (Figure 5c). The impaired PFS for the group of patients with ≥4 cell CTC-cluster remained significant in multivariable analyses, adjusting for clinicopathological factors (Table 5). Similar findings were observed when analyzing the risk of death; patients with 2–3 cell CTC-cluster had a HR of 7.25 (95% CI 2.04–25.7; *p* = 0.002), and patients with ≥4 cell CTC-cluster had a HR of 12.5 (95% CI 2.5–63.2; *p* = 0.002), when compared with patients without CTCs. The mean OS time for patients with 2–3 cell CTC-cluster and ≥4 cell CTC-cluster was of 49 and 35 days (Figure 5d). Again, the effect observed in patients with ≥4 cell CTC-cluster remained significant after adjusting for other clinicopathological variables (Table 5).

In addition, to better understand the clinical significance of CTC-clusters in this cohort of patients we look at the maintained presence of CTC-clusters in the blood of patients across the two time points in relation to PFS and OS. Among patients with ≥ 1 CTC (*n* = 45), 2 (4.4%) had CTC-clusters across two time points, and 16 (35.6%) had CTC-clusters at one time point, whereas 27 (60.0%) had only single CTCs (Figure 5e). Regarding PFS, patients with CTC-clusters had a shorter mean PFS time than patients with only single CTCs (*P*_log-rank_ = 0.017; Figure 5f). However, only patients with CTC-clusters at one time point had a significant increase on risk of progression (HR_PFS_ 2.8; 95% CI 1.22–6.34; *p* = 0.014). Similarly, patients with CTC-clusters across one and two time points had a shorter mean OS compared to patients with only single CTCs (*P*_log-rank_ = 0.007; Figure 5g). Both groups of patients had a significantly higher risk of death (HR_OS_ 3.3; 95% CI 1.29–8.6; *p =* 0.012, and HR_OS_ 6.5; 95% CI 1.31–32.7; *p* = 0.021, respectively). Our data suggest that the continuous presence of CTC-clusters in the blood of MBC is associated with a shorter survival.

## 3. Discussion

The presence of CTC-clusters can be detected in the blood of patients with various tumors, including BC [16,17,18,19,20,21]. Evidences for the role of CTC-clusters in the development of metastasis are building up, and both the cellular and molecular mechanisms are now being revealed. Indeed, studies have shown that CTC-clusters hold an increased metastatic potential [22,23,32,33], and have a survival advantage over single CTCs [16,17,22]. Moreover, their hybrid epithelial-mesenchymal phenotype [26,34] together with the stem cell features and the enrichment on genes related to proliferation and DNA replication [24,25], seem to be some of the underlying reasons for their enhanced metastatic potential. Furthermore, in addition to the well supported prognostic value of CTC counts [5,9,10,11,12] and CTC-clusters [16,22,27,28,29,30] in MBC, clinical studies have shown that the detection of CTC-clusters adds prognostic value to CTC enumeration alone [27,28,29,30]. All of this evidence goes in support of a relevant biological implication of CTC-clusters in the progression of BC towards a metastatic disease.

It is important to remember that most of the evidences of the clinical impact of CTC-clusters in BC have been gathered from prospectively designed clinical studies on rather homogeneous and well selected cohorts of metastatic patients [16,27,29,30]. The aim of our study, however, was to evaluate the prognostic value of CTC-clusters in a heterogeneous and small real-world cohort of MBC patients in order to determine whether we could reproduce some of the data reported in the clinical trials and whether these data could be useful in the clinic. Our study focused on an unselected population of patients diagnosed with MBC starting first line of therapy or initiating a new line of therapy (up to fourth line of therapy). Therefore, the importance of our study relies on the fact that, in despite of these limitations, clear evidences on the prognostic value of CTC-clusters are observed.

In this study, assessing the prognostic value of CTCs and CTC-clusters in a cohort of MBC patients, our results validate data from previous studies and show that the presence of high CTC counts in the blood of patients at baseline predicts a poorer prognosis in terms of OS, and that the presence of CTC-clusters predicts shorter progression and survival times. At follow up, high CTC counts predict shorter survival time, whereas the presence of CTC-clusters does not show prognostic value. We also observed that patients with CTC count ≥ 5 were those with a higher frequency of CTC-clusters, evidencing a combined dissemination pattern, as was previously described [28]. Moreover, we show that the presence of CTC-clusters at baseline is a strong independent prognostic factor of patient outcome. In addition, the longitudinal analysis of CTC and CTC-cluster counts during the course of disease evidenced that CTC-cluster enumeration adds significant prognostic value to CTC enumeration alone, and that CTC-cluster size correlates with patient OS. Lastly, our data corroborate that the continuous presence of CTC-clusters in the blood of MBC patients is associated with shortened patient survival.

Several studies over the last fifteen years, including a recent pooled analysis of 20 independent studies with data from 1944 patients [5], have demonstrated the prognostic value of CTCs in patients with MBC. These studies showed that a CTC count of ≥5 CTCs per 7.5 ml blood is an independent predictor of worse PFS and OS. Our study partially reproduces these data, as we did not observe significant differences in PFS between patients with ≥ 5 CTCs and patients with <5 CTCs neither at baseline, nor at follow up, probably because of the heterogeneous cohort analyzed. Regarding the clinical validity of CTC-clusters, previous studies have shown the prognostic value of baseline CTC-clusters in patients with MBC [27,28,30]. However, despite all these evidences, a very recently published study showed that, compared with enumeration of CTCs alone, the presence of CTC-clusters was not prognostic in patients who had ≥5 CTCs/7.5 ml whole blood at baseline [31]. This study suggests that CTC-clusters do not play a major role in the outcome of MBC patients starting first line chemotherapy, and that mortality depends on the number of CTCs. Our data suggest that the presence of CTC-clusters in patients with ≥20 CTCs may add prognostic value to CTC enumeration alone when compared to patients with <20 CTCs. However, CTC-cluster were not independent of CTC counts when applying the cutoff ≥20 CTCs/7.5 mL whole blood. Nevertheless, our data in this regard have to be carefully interpreted, and a larger cohort of patients would be needed to appropriately address the additional prognostic value of CTC-clusters in patients with high CTC counts (≥20 CTCs/7.5 mL).

Our data support a role for CTC-clusters as a prognostic factor at baseline in MBC, but they are not consistent with findings showing that CTC-cluster levels measured at follow up are also associated with patient outcomes [16,28,30]. In order to understand these discrepancies, some limitations of our study need to be considered. As previously mentioned, our study cohort is a mixed population of MBC patients, including patients starting first line of therapy and also patients initiating a new line of therapy. Of the 54 patients considered, 44 started with the first line of therapy and among these, CTCs ≥5 and CTC-clusters ≥1 were detected in 23 (52%) and 7 (16%) patients, respectively. Moreover, follow up samples were collected from 38 out of the 54 patients enrolled, limiting therefore the statistical power of the analysis. Of them, 13 patients (34.2%) had ≥5 CTCs and only 7 (18.4%) had detectable CTC-clusters.

We also analyzed in our cohort of patients whether the presence of CTC-WBC clusters could predict patient outcomes. Jansson et al. [29] previously reported that patients with WBC-CTCs have a worse prognosis, although this association was only found after six months of initiation of systemic therapy. Of note, we did not observe any association between the presence of clustered CTC-WBC and PFS or OS, neither at baseline nor at follow up. A possible explanation for this lack of association is the restricted number of patients from our cohort in whom CTC-WBC were detected; two patients at baseline and three at follow up. Another feasible explanation is the short follow up time in our cohort compared to the cohort presented by Jansson et al. Nevertheless, the close relationship between CTCs and cells from the immune system grants further investigation, particularly under the light of recent findings [35,36,37]. This may be particularly relevant after the results reported by Szczerba et al. and Sprouse et al., showing that the interaction of neutrophils and PMN-MDSCs with CTCs enhances their metastatic potential [35,37].

Few studies have reported the independent prognostic value of CTC-cluster enumeration and the added prognostic value of CTC-cluster to CTC enumeration alone, both at baseline and follow up [16,27,28,30]. In keeping with this, our data showed that combining CTC and CTC-cluster counts allows for the stratification of patients in risk groups. Thus, patients with ≥5 CTCs with CTC-clusters at baseline had a shorter PFS and OS than patients with <5 CTC without CTC-clusters. A similar result on OS was observed during follow up. These data indicate that in this cohort of patients, baseline CTC-clusters are an independent prognostic factor adding value to the enumeration of CTCs alone, while this association is less strong at follow up. The weaker predictive value observed in the follow up group was due to the limited statistical power of the analysis influenced by the low number of patients from whom follow up sample was collected, as well as the short follow up time of some patients in our cohort. 

Recently, the prognostic value of changes in CTC and CTC-cluster counts has been investigated, along the treatment of the disease, from baseline to follow up. A prospective study conducted by Larsson et al. in a cohort of newly diagnosed MBC patients starting first-line systemic therapy showed that a reduction in CTC count from ≥5 CTCs at baseline to <5 CTCs at follow up times (measured one, three, and six months after therapy initiation) was significantly associated with improved survival [30]. Moreover, they also showed that the presence of CTC-clusters in the blood of patients during systemic treatment increased the risk of progression and death. In agreement with these data, the study by Wang et al. on a cohort of MBC patients enrolled when starting a new therapy showed that both changes of CTCs and CTC-clusters predicted patient outcomes in terms of PFS and OS [28]. Taken together, these studies suggest the importance of monitoring early changes in CTCs and CTC-clusters in response to treatment. In agreement with this, we found that patients with ≥ 5 CTCs at both time points had a significantly shorter median OS than those patients with low CTC counts, indicating that the reduction of CTC counts during treatment is predictive of increased patient survival. This result is also consistent with a recent meta-analysis showing that CTC status is a useful indicator to monitor the treatment response and to predict the outcome of patients with MBC [13]. The analysis based on CTC-clusters however, showed that changes in CTC-clusters from baseline to follow up failed to predict patient progression and survival, although a trend towards a shorter survival was observed for patients classified as being in the high-risk group. The lack of predictive value of CTC-clusters in our study is possibly due to the number of patients with follow up data (only 18.4% of patients at follow up presented detectable CTC-clusters), together with the single follow up time point measured in our study, that might weakened the prognostic value of CTCs and CTC-clusters, as pointed out by a previous report [28]. However, because patients in the high-risk group showed a non-significant association with a worse outcome, we cannot rule out the importance of monitoring early changes in CTC-clusters in response to treatment. Nevertheless, the capacity of CTC enumeration to guide therapy is limited, and deeper biological and molecular knowledge of these cells is needed.

Interestingly, our longitudinal study analyzing a total of 96 samples collected across the two time points indicates that CTC-clusters have a prognostic effect and that they provide additional independent prognostic value to CTCs. In addition, we also showed that the presence of CTC-clusters across both time points was predictive of a much shorter survival time, although this result should be evaluated in a larger cohort, with adequate power for the analysis. Altogether, our data are in line with previous reports [22,28] and suggest that the continuous presence of CTC-clusters in the blood of MBC is associated with a worse outcome, and that the longitudinal monitoring of CTC-clusters might increase their prognostic value.

A novel link between the size of CTC-clusters and the survival of MBC patients has been reported [28]. Wang et al. showed that patients with 2-cell and ≥ 3-cell CTC-clusters had a higher risk of death than patients without any CTC. Moreover, preclinical studies have already shown that the success rate of CTC-clusters on the formation of metastases partially depends on the size and the concentration of the clusters found in the blood [38,39,40,41]. Our data are consistent with these findings, and show that, in relation to OS, patients with 2–3 cell CTC-cluster and patients with ≥4 cell CTC-cluster were at a higher risk of death compared to patients without any CTC. In addition, our findings also suggest a possible association with disease progression, not yet reported, as patients with ≥4 cell CTC-clusters were at a higher risk of progression. However, these data have to be valued with caution, as the power was limited, due to the small number of patients presenting CTC-clusters composed by ≥4 cells. Taken together, and despite the small number of patients with CTC-clusters in our cohort, these data indicate that CTC-cluster counts, but also CTC-cluster size, may be important in patient prognostication. Moreover, they suggest that a larger cluster size might confer a biological advantage during disease progression (i.e. increased survival and/or proliferation capacity of CTCs). Indeed, studies have shown the extremely low incidence of apoptotic CTCs within CTC-clusters, in contrast to single CTCs, as well as an enrichment on genes related to cell proliferation in CTC-clusters [16,17,25,29]. These observations may have relevant therapeutic implications and CTC-clusters may become a target in BC. In this sense, preclinical studies suggest that the pharmacological targeting of CTC-cluster size could be a strategy to suppress metastasis formation in BC [25], a strategy that is currently under evaluation in a Phase 1 clinical trial (NCT03928210).

## 4. Materials and Methods 

### 4.1. Study Subjects 

Female patients diagnosed with MBC at the Clinical Hospital of Santiago de Compostela (Spain) were recruited for the study from March 2014 to September 2018. The last follow up date was 30 September 2018. All patients involved in the study signed an informed consent to participate, approved by the Clinical Research Ethics Committee (CEIC) of Galicia (code 2013/462 or 2015/772). Blood samples used for the enumeration of CTCs and CTC-clusters were collected from patients at baseline, and during the course of treatment as a follow up sample. Baseline was defined as the time of sample collection from patients who were about to start first-line of systemic therapy or patients who were about to start a new line of therapy. Follow up samples were only collected from patients who started the first line of systemic therapy, and were commonly followed for 3–5 weeks after treatment initiation; however, the follow up time varied among patients, as determined by the physicians.

### 4.2. Enumerations of CTCs and CTC-Clusters 

Approximately 7.5 ml of whole blood was collected into CellSave Preservative Tube containing a cellular fixative (Menarini Silicon Biosystems, Bologna, Italy), maintained at room temperature and processes within the following 96 h upon collection at the Liquid Biopsy Analysis Unit of the Health Research Institute of Santiago de Compostela (IDIS). Isolation and enumerations of CTCs and CTC-clusters were performed using the CellSearch^®^ System (Menarini Silicon Biosystems), using the CellSearch Epithelial Circulating Tumor Cell Kit. Blood samples were enriched for CTCs using ferrous particles coated with an antibody, recognizing the epithelial cell adhesion molecule (EpCAM) antigen. Enriched cells were then stained with fluorescent antibodies against cytokeratins (CKs) 8, 18, and 19, CD45, and the double-stranded DNA stain DAPI, and scanned with the microscope CellTracks Analyzer II (Menarini Silicon Biosystems). Cells were presented in a gallery for manual evaluation by two trained technicians. CTCs were identified by round-oval morphology, positive staining for PE-conjugated CKs and DAPI, and negative staining for APC-conjugated CD45 antibodies, following the criteria previously described for the CellSearch system. CTC-clusters were defined as groups of two or more individual CTCs (CKs-PE and DAPI positive and CD45-APC negative staining), with distinct non-overlapping nuclei and intact cytoplasm membranes. No additional staining of CTCs was performed after the CellSearch analysis. In addition, the interaction of CTCs with white blood cells (WBC), defined by CD45-APC positive staining and DAPI, was evaluated.

### 4.3. Statistical Analysis 

Statistical power calculations were done for two groups (<5 CTC and ≥5 CTCs) based on the size of the groups, 23 and 31 patients respectively; hazard ratio 0.31 of <5 CTC relative to ≥5 CTCs, accrual period of 1624 days, and alpha level 0.05. The resulting power analysis was 0.887. The program PS, Power and Sample Size Calculation version 3.1.6, 2018 was used for these calculations. The endpoints analyzed in this study were progression-free survival (PFS) and overall survival (OS) in relation to the number of CTCs and CTC-clusters. PFS was defined as the time from blood sample collection to the date of clinical progression or death; and OS was defined as the time from blood collection to the date of death. Patients without an endpoint event at follow up were censored. Survival analysis was performed using Kaplan–Meier method and curves compared using log-rank test. Associations between CTC and CTC-cluster counts with PFS and OS were estimated using hazard ratios (HR) and 95% confidence intervals (CI), calculated by univariate and multivariate Cox proportional hazards models. Multivariable analyses were adjusted for age, ECOG, BC subtype, site of metastasis, number of metastatic sites and treatments. CTC characteristics, i.e. CTC-clusters and CTC-WBC cluster, across breast cancers subtypes and at different time-points were compared using a Pearson Chi-squared test. Data analyses were performed in the “R” open source software environment (version 3.4.4, https://www.r-project.org/) and GraphPad Prism (version 6.01, https://www.graphpad.com/). 

## 5. Conclusions

The evaluation of the presence of CTC-clusters in a heterogeneous and small real-world cohort of MBC patients supports that the enumeration of CTC-clusters at baseline is a strong independent predictor of PFS and OS in MBC patients, and that the longitudinal monitoring of CTC-clusters might increase their prognostic value. Moreover, our analysis shows that patients with CTC-cluster of a larger size are at a higher risk of death, and that the persistent presence of CTC-cluster in the blood of patients during the course of treatment is also a predictor of a worse patient outcome. These findings in our real-world cohort of patients also support the biological significance of CTC-clusters in tumor progression.

## Figures and Tables

**Figure 1 cancers-12-01111-f001:**
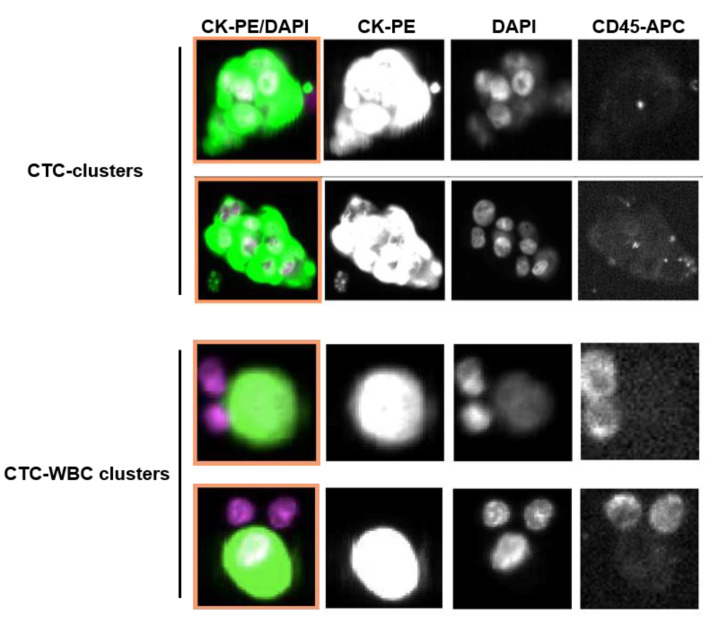
Representative images of circulating tumor cell (CTC)-clusters and CTC-WBC (white blood cell) clusters isolated from 7.5 ml of blood from metastatic breast cancer (MBC) patients captured by CellSearch® system. CTCs were identified by round-oval morphology, positive staining for cytokeratins 8, 18, and 19 (CKs-PE, phycoerythrin-conjugated antibody) and DAPI (DNA dye), and negative staining for CD45 (CD45-APC, allophycocyanin-conjugated antibody.

**Figure 2 cancers-12-01111-f002:**
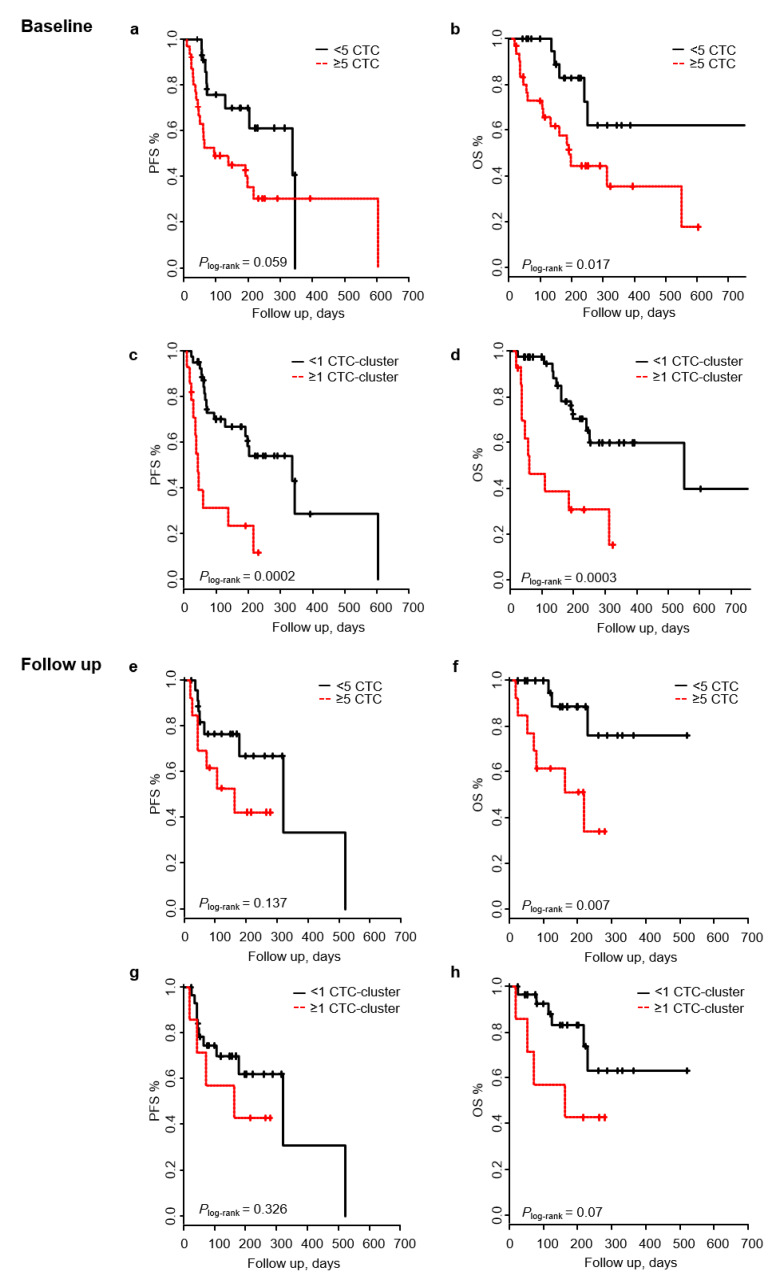
CTCs and CTC-clusters at baseline and follow up, in relation to progression-free survival (PFS) and overall survival (OS). Kaplan–Meier curves displaying PFS and OS at baseline based on CTC count (≥5 CTCs/7.5 ml of blood) (**a**,**b**) and CTC-cluster count (≥1 CTC-cluster/7.5 mL of blood) (**c**,**d**), and at follow up based on CTC count (**e**,**f**) and CTC-cluster count (**g**,**h**).

**Figure 3 cancers-12-01111-f003:**
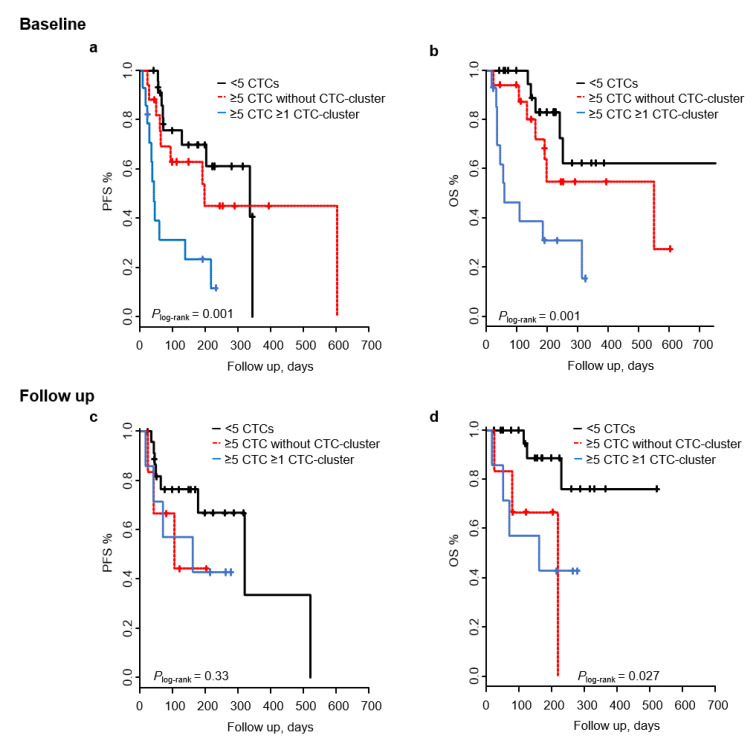
Risk groups regarding PFS and OS based on CTC count and CTC-clusters. Kaplan–Meier curves displaying PFS and OS of patients with <5 CTC, patients with ≥5 CTCs without CTC-clusters and patients ≥5 CTCs with CTC-clusters at baseline (**a**,**b**) and follow up (**c**,**d**).

**Figure 4 cancers-12-01111-f004:**
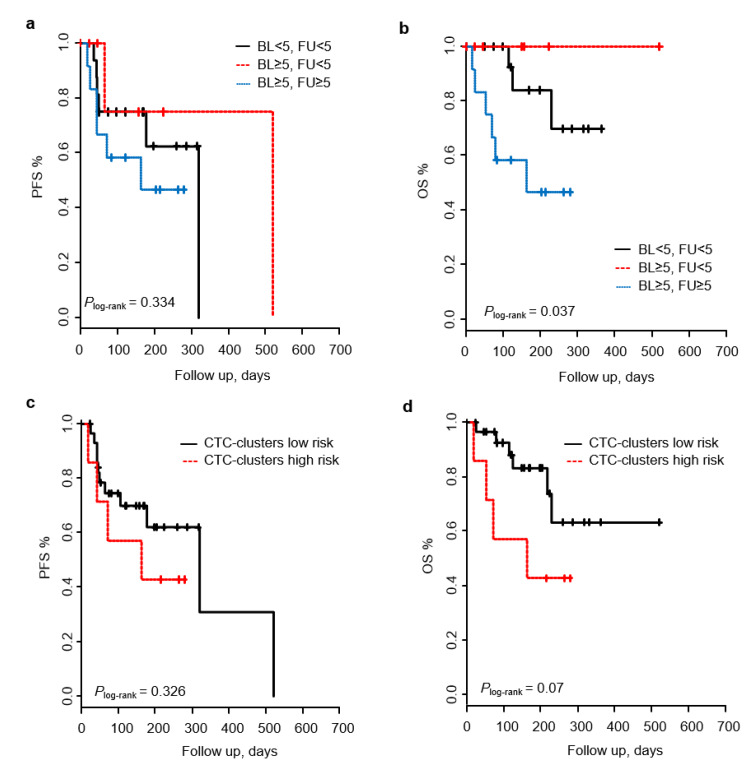
Baseline-to-follow up changes in CTCs and CTC-clusters in relation to patient outcome. Based on CTC counts, Kaplan–Meier curves displaying PFS and OS for the three risk group established: <5 CTCs both at baseline (BL) and follow up (FU); ≥5 CTCs at BL and <5 CTCs at FU; and ≥5 CTCs both at BL and FU (**a**,**b**). Based on CTC-cluster counts, Kaplan–Meier curves displaying PFS and OS for the two risk groups established, patients without CTC-clusters at both time points or with a reduction in the number from BL to FU (low risk); and patients whose CTC-cluster count did not change or increase over time (high risk) (**c**,**d**).

**Figure 5 cancers-12-01111-f005:**
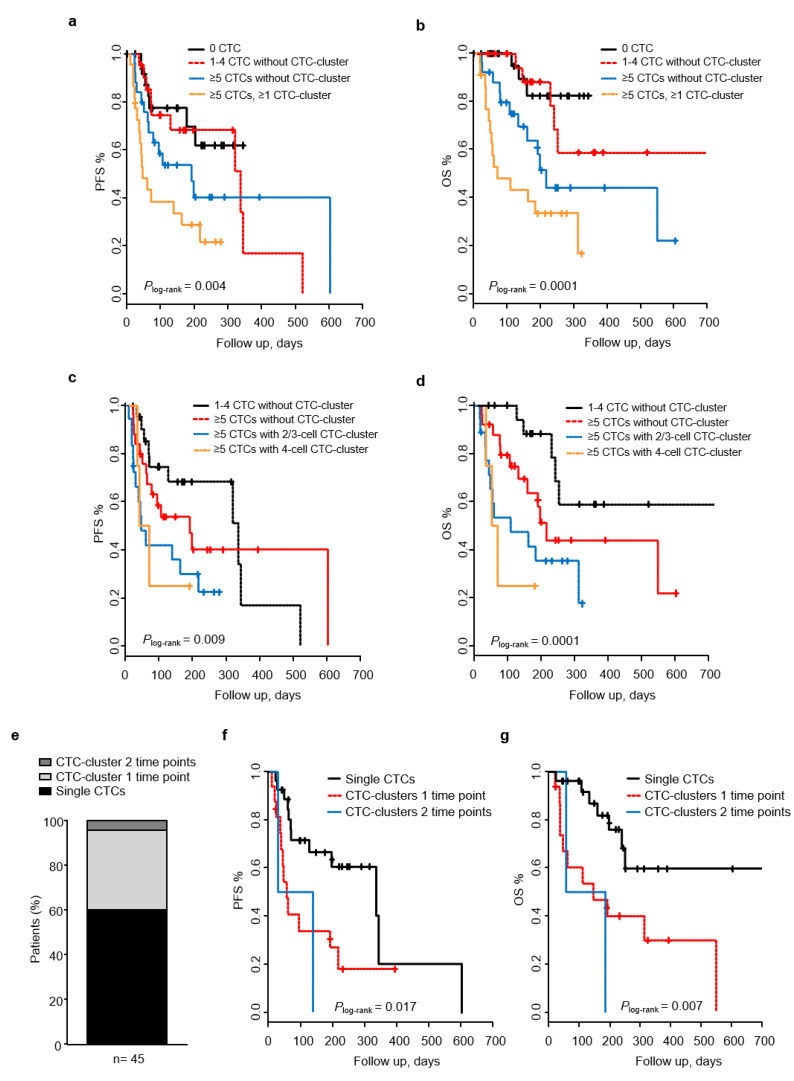
Longitudinal analysis for CTCs and CTC-clusters in relation to patient outcome. Kaplan–Meier curves displaying PFS and OS of patients without CTCs, patients with 1–4 CTC without CTC-clusters, patients with ≥5 CTCs without CTC-clusters and patients ≥5 CTCs with CTC-clusters (**a**,**b**). The predictive value of CTC-cluster size was also analyzed. Kaplan–Meier curves displaying PFS and OS of patients with 1–4 CTC without CTC-clusters, patients with ≥5 CTCs without CTC-clusters, patients ≥5 CTCs with 2–3 cell CTC-clusters, and patients ≥ 5 CTCs with 4 cell CTC-clusters (**c**,**d**). Bar graph showing the percentage of patients positive for CTCs having single CTCs only (black), CTC-clusters during one time point (light gray), and CTC-clusters across two time points (dark grey) (**e**). Kaplan–Meier curves displaying PFS and OS of patients with single CTCs only, patients with CTC-clusters during one time point, and patients with CTC-clusters across two time points (**f**,**g**).

**Table 1 cancers-12-01111-t001:** Patient characteristics (*n* = 54).

Variables	Total, *n* (%)
Age (years), mean	58.5
<65 years	34 (63.0)
≥65 years	20 (37.0)
Tumor stage	
IV	54 (100.0)
Baseline ECOG	
0	18 (33.3)
1	31 (57.4)
2	5 (9.3)
Primary tumor NHG	
I	4 (7.4)
II	26 (48.1)
II	15 (27.8)
Unknown	9 (16.7)
Breast cancer subtypes	
HR+HER2-	27 (50.9)
HER2+	11 (20.8)
HR-HER2-	15 (25.8)
Unknown	1 (1.8)
Site of metastasis *	
Non-visceral	14 (25.9)
Visceral	40 (74.1)
Number of metastatic sites	
<3	29 (53.7)
≥3	25 (46.3)
First-line systemic therapy (*n* = 44)	
Chemotherapy	23 (52.3)
Hormonal therapy	15 (34.1)
Target therapy ^#^	6 (13.6)
Other lines (*n* = 10)	
Chemotherapy	8 (80.0)
Hormonal therapy	1 (10.0)
Target therapy ^#^	1 (10.0)

Abbreviations: ECOG Eastern Cooperative Oncology Group, NHG Nottingham histologic grade, HR hormone receptor, HER2 human epidermal growth factor receptor 2, TNBC triple negative breast cancer. * Visceral metastases were defined as lung, liver, peritoneal, and/or pleural involvement and non-visceral metastasis was defined as lymph node and/or bone involvement. ^#^ In combination with chemotherapy.

**Table 2 cancers-12-01111-t002:** Associations between CTCs and CTC-clusters and PFS and OS of MBC patients.

Variable	Total	Events, *n* (%)	HR (95% CI)	*p* Value	HR (95% CI) *	*p* Value *
Baseline						
Associated with PFS						
CTCs						
<5	23	11 (47.8)	1.00		1.00	
≥5	31	20 (64.5)	2.11 (0.95–4.70)	0.0649	2.82 (1.15–6.87)	0.022
CTC-clusters						
No	40	18 (45.0)	1.00		1.00	
Yes	14	11 (78.6)	3.95 (1.80–8.68)	0.0006	4.46 (1.55–12.8)	0.005
Associated with OS						
CTCs						
<5	23	6 (26.1)	1.00		1.00	
≥5	31	17 (54.8)	3.15 (1.16–8.55)	0.024	3.33 (1.14–9.73)	0.027
CTC-clusters						
No	40	13 (32.5)	1.00		1.00	
Yes	14	10 (71.4)	4.23 (1.8–10.1)	0.0009	6.55 (1.78–23.8)	0.004
Follow up						
Associated with PFS						
CTCs						
<5	25	8 (32)	1.00		1.00	
≥5	13	7 (53.8)	2.3 (0.76–6.7)	0.15	2.39 (0.47–12.04)	0.29
CTC-clusters						
No	31	11 (35.5)	1.00		1.00	
Yes	7	4 (57.1)	1.8 (0.55–5.9)	0.33	5.05 (1.24–20.52)	0.023
Associated with OS						
CTCs						
<5	25	3 (12.0)	1.00		1.00	
≥5	13	7 (53.8)	5.3 (1.4–21)	0.017	9.6 × 10^17^ (0.0-Inf)	0.996
CTC-clusters						
No	31	6 (19.4)	1.00		1.00	
Yes	7	4 (57.1)	3 (0.83–11)	0.09	29.74 (2.55–345.8)	0.006

Abbreviations: CTC circulating tumor cell, PFS progression-free survival, OS overall survival, HR hazard ratio, CI confidence interval, Inf infinity. * Adjusted for age, ECOG, subtype, number of metastatic sites, site of metastasis and treatments.

**Table 3 cancers-12-01111-t003:** Joint effect analysis of CTCs and CTC-clusters with patient PFS and OS.

Variables	Total	Events, *n* (%)	HR (95% CI)	*p* Value	HR (95% CI) *	*p* Value *
Joint effect of baseline CTC and CTC-clusters						
Associated with PFS						
<5 CTC without CTC-cluster	23	9 (39.13)	1.00		1.00	
≥5 CTC without CTC-cluster	17	9 (52.84)	1.24 (0.47–3.24)	0.66	1.74 (0.57–5.30)	0.32
≥5 CTC, ≥1 CTC-cluster	14	11 (78.57)	4.34 (1.76–10.6)	0.0013	5.16 (1.68–15.8)	0.0041
Associated with OS						
<5 CTC without CTC-cluster	23	6 (26.1)	1.00		1.00	
≥5 CTC without CTC-cluster	17	7 (41.18)	1.88 (0.59–5.98)	0.28	1.84 (0.50–6.82)	0.36
≥5 CTC, ≥1 CTC-cluster	14	10 (71.43)	5.79 (1.96–17.1)	0.0014	7.79 (1.93–31.4)	0.0038
Joint effect of first follow up CTC and CTC-clusters						
Associated with PFS						
<5 CTC without CTC-cluster	25	8 (32.0)	1.00		1	
≥5 CTC without CTC-cluster	6	3 (50.0)	2.3 (0.57–9.27)	0.24	0.61 (0.08–4.53)	0.635
≥5 CTC, ≥1 CTC-cluster	7	4 (57.1)	2.2 (0.62–7.84)	0.22	4.45 (1.01–19.59)	0.0484
Associated with OS						
<5 CTC without CTC-cluster	25	3 (12.0)	1.00		1.00	
≥5 CTC without CTC-cluster	6	3 (50)	5.71 (1.11–29.4)	0.037	1.2 × 10^18^ (0.0-Inf)	0.996
≥5 CTC, ≥1 CTC-cluster	7	4 (57.1)	5.00 (1.11–22.4)	0.0353	2.4 × 10^18^ (0.0-Inf)	0.996

Abbreviations: CTC circulating tumor cell, PFS progression-free survival, OS overall survival, HR hazard ratio, CI confidence interval, Inf infinity. * Adjusted for age, ECOG, subtype, number of metastatic sites, site of metastasis and treatments.

**Table 4 cancers-12-01111-t004:** Change in CTC and CTC-cluster count from baseline to follow up.

Variables	Total	Events, *n* (%)	HR (95% CI)	*p* Value	HR (95% CI) *	*p* Value *
Baseline CTC to Follow up CTC						
Associated with PFS						
<5 CTC, <5 CTC	18	6 (33.3)	1.00			
≥5 CTC, <5 CTC	7	2 (28.6)	0.430 (0.05–3.70)	0.443	15.25 (0.42–553.3)	0.137
≥5 CTC, ≥5 CTC	12	6 (50)	1.82 (0.572–5.82)	0.308	3.09 (0.51–18.64)	0.217
Associated with OS						
<5 CTC, <5 CTC	18	3 (16.7)	1.00			
≥5 CTC, <5 CTC	7	0 (0.0)	1.67 × 10^−8^ (0.0-Inf)	0.998	1.45 × 10^9^ (0.0-Inf)	0.997
≥5 CTC, ≥5 CTC	12	6 (33.3)	3.91 (0.97–15.79)	0.055	9.75 × 10^17^ (0.0-Inf)	0.999
Baseline CTC-cluster to Follow up CTC-cluster						
Associated with PFS						
Low risk	31	11 (35.5)	1.00		1.00	
High risk	7	4 (57.1)	1.79 (0.55–5.85)	0.33	5.05 (1.245–20.52)	0.023
Associated with OS						
Low risk	31	6 (19.4)	1.00		1.00	
High risk	7	4 (57.1)	2.96 (0.83–10.56)	0.09	29.7 (2.55–345.8)	0.006

Abbreviations: CTC circulating tumor cell, PFS progression-free survival, OS overall survival, HR hazard ratio, CI confidence interval, Inf infinity. * Adjusted for age, ECOG, subtype, number of metastatic sites, site of metastasis and treatments.

**Table 5 cancers-12-01111-t005:** Longitudinal changes in CTCs and CTC-clusters in relation to PFS and OS.

Variable	Total	Events, *n* (%)	HR (95% CI)	*p* Value	HR (95% CI) *	*p* Value *
Associated with PFS						
No CTC	28	7 (25.0)	1.00		1.00	
1–4 CTC without CTC-cluster	21	10 (47.6)	1.39 (0.52–3.73)	0.50	1.85 (0.62–5.46)	0.26
≥5 CTC without CTC-cluster	25	14 (56.0)	1.82 (0.72–4.62)	0.20	1.87 (0.69–5.05)	0.21
≥5 CTC with 2–3 cell CTC-cluster	18	13 (72.2)	3.97 (1.58–9.97)	0.0033	1.85 (0.34–10.0)	0.470
≥5 CTC with ≥4 cell CTC-cluster	4	3 (75.0)	4.10 (1.05–16.0)	0.0416	6.94 (1.32–36.4)	0.021
Associated with OS						
No CTC	28	3 (10.7)	1.00		1.00	
1–4 CTC without CTC-cluster	21	6 (28.7)	1.59 (0.38–6.70)	0.52	2.62 (0.51–13.3)	0.24
≥5 CTC without CTC-cluster	25	12 (48.0)	4.02 (1.12–14.38)	0.0321	6.05 (1.47–24.8)	0.01
≥5 CTC with 2–3 cell CTC-cluster	18	12 (78.6)	7.25 (2.04–25.74)	0.0021	1.49 (0.13–16.2)	0.741
≥5 CTC with ≥4 cell CTC-cluster	4	3 (75)	12.55 (2.49–63.2)	0.0021	73.73 (8.77–619.4)	0.00007

Abbreviations: CTC circulating tumor cell, PFS progression-free survival, OS overall survival, HR hazard ratio, CI confidence interval. * Adjusted for age, ECOG, subtype, number of metastatic sites, site of metastasis and treatments.

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
