# Peer review of "Analysis of a Real-World Cohort of Metastatic Breast Cancer Patients Shows Circulating Tumor Cell Clusters (CTC-clusters) as Predictors of Patient Outcomes"

_cancers, 2020, doi:10.3390/cancers12051111_

Round 1

Reviewer 1 Report

The paper present a population of 54 breast cancer patients, which they describe as quite heterogenous (i would like to have a table showing patient characteristics for a quick overview in the article). The number of CTC were measured at two timepoints as were the number of clusters. Their prognostic value were evaluated. It seems CTC were prognostic, as were clusters, with possibly the combination being even stronger associated with worse outcomes. 

The study is one in a longer line and should be improved with more novel outcomes, especially considering a relatively small number of included patients. While i normally would not consider a small number a large problem, the issue is that there is no power analysis. And many analyses have been performed with less than 10 patients in several groups, making results unreliable, especially of the more interesting outcomes: Do clusters have ADDITIONALL value to CTC alone.

specific notes

  • why do you call it a retrospective study, considering all the data was prospectively gathered? (blood samples etc, selection of the cohort). 
  • in the introduction it is stated as fact that clusters outperform CTC< but I am not so sure this is the case. However, in the MBC field i am less well versed, so could be wrong
  • power analysis needs to be added
  • novelty: prognosis of CTC/clusters is known, and the CTC with clusters interaction has been explored as well. The population is too small for reliable results in this regard too. Possibly compare CTC/clusters with PET-CT imaging, which has been performed before as well, but less times?
  • Several times the heterogeneity is mentioned as a boon. Why would it influence your results? And why in this way? And how would the different treatments influence the survival? Are these skewed or not? to my knowledge CTC counts are not influenced by therapy or other patient characteristics, except possibly vessel invasion of the primary tumor in early stages and CTC are prognostic in all cases, so why would this influence your results in any way?
  • Correlation between clusters and CTC have been made visible in one figure. But if the interaction is one of the major (and innovative) results, this point should be highlighted more
  • Figure S1 is difficult to read in this format. Maybe present is horizontally?

Reviewer 2 Report

The authors report evaluation of prognostic value of CTC-clusters in 54 MBC patients. The authors have done a good job in retrospective analysis of the data. Please address the following concerns

  1. Figure S3. In the CTC-clusters there is signal in CD45_APC. Isnt cluster should be negative for CD45? If this noise, the signal seems comparable to CTC-WBC cluster. How is the signal to noise estimated?
  2. The Cell search system uses a EpCAM based enrichment. It is well known that a major subset of CTCs in BC don't express EpCAM. What are the limitations of this study with regards to this.
  3. Line 80 highlights a study that absolute CTCs are better predictor than CTC-cluster. Please contrast how this study is not in agreement with the data presented here.

Reviewer 3 Report

This manuscript describes that patients with high circulating tumor cell (CTC) counts and CTC-cluster were at a higher risk of progression and death, and patients with CTC-clusters had shorter survival compared to patients without clusters.

In the title, “CTC” should be “circulating tumor cell”.

In the first line of the Abstract, “CTC enumeration” should be “circulating tumor cell (CTC) enumeration”.

The definition of “baseline” should be described.

In the most figures, the follow-up days are different among the cohorts. The data could be cut down for matching the follow-up days in each figure.

Reviewer 4 Report

This manuscript was prepared very well with nicely written and structured. Although CTC and CTC-clusters have been mentioned as a prognostic predictor in cancer, this study still contributes to this field with many clinical evidence.

Only some suggestions were provided:

In Materials and Methods, the procedure of CTC purification, including who to exclude non-CTC population, and the definition of CTC-clusters should have to been described with more detail information. In addition, it is also lack of experimental procedure for immunostaining of CTC. 

The characteristics of CTC by immunostaining (Figure S3) should be shown in the main text. The quality of this experiment needs improvement.

Round 2

Reviewer 1 Report

In general i feel like most aspects that i have seen have been adressed The atuhors make a compelling point about the novelty of their study and the additionall value it has to the growing base of eveidence. Still I feel that there are some small adjustments that could be implemented. Mosly regarding the finding about clusters and CTC values and their association with sruvial before and after treatment.

  • In the survival curves at follow up, a large number of patients in the group without CTC have been censored after a relatively short follow up, making survival curves less reliable, possibly the survival coul be updated 
  • It would be good for the authors to focus a bit more on the difference in survival between patients with CTC without clusters and those with clusters and CTC. Survival there seems completely determined by clusters. Which is strange, as clusters and CTC lines at follow up are not diverging at all. It could be that this is an artefact due to low numbers included, and that we are actually looking at the response to therapy. However, the fact that clusters are associated with low response rates would also be interesting. I would be interested in seeing a bit more about this in the discussion
  • baseline in the methods should be more clearly defined: is it always before new treatment has been started? how many weeks or days beforre start of treatment were acceptable? Did all patients in the follow up group continue with the Original treatment?
  • I feel that supplementary table S1 should be part of the article, not supplementary as it highlight the study population.
  • The use of 20 CTC as a cut off value is strange, considering the official one is 5. But i assume it is requested by a different reviewer.
